# Integrated Optical Waveguide Electric Field Sensors Based on Bismuth Germanate

**DOI:** 10.3390/s24175570

**Published:** 2024-08-28

**Authors:** Jin Wang, Yilin Song, Xuefei Song, Wei Zhang, Junqi Yang, Zhi Xuan

**Affiliations:** School of Mechanical Engineering and Electronic Information, China University of Geosciences, Wuhan 430074, China; song10@cug.edu.cn (Y.S.); songxuefei0123@cug.edu.cn (X.S.); zhangwei0224@cug.edu.cn (W.Z.); yangjunqi@cug.edu.cn (J.Y.); xuanzhi2001@cug.edu.cn (Z.X.)

**Keywords:** bismuth germanate, optical electric field sensor, femtosecond laser writing

## Abstract

Bismuth germanate (Bi_4_Ge_3_O_12_, BGO) is a widely used optical sensing material with a high electro-optic coefficient, ideal for optical electric field sensors. Achieving high precision in electric field sensing requires fabricating optical waveguides on BGO. Traditional waveguide writing methods face challenges with this material. This study explores using femtosecond laser writing technology for preparing waveguides on BGO, leveraging ultrafast optical fields for superior material modification. Our experimental analysis shows that a cladding-type waveguide, written with a femtosecond laser at 200 kHz repetition frequency and 10.15 mW average power (pulse energy of 50.8 nJ), exhibits excellent light-guiding characteristics. Simulations of near-field optical intensity distribution and refractive index variations using the refractive index reconstruction method demonstrate that the refractive index modulation ensures single-mode transmission and effectively confines light to the core layer. In situ refractive index characterization confirms the feasibility of fabricating a waveguide with a refractive index reduction on BGO. The resulting waveguide has a loss per unit length of approximately 1.2 dB/cm, marking a successful fabrication. Additionally, we design an antenna electrode, analyze sensor performance indicators, and integrate a preparation process plan for the antenna electrode. This achievement establishes a solid experimental foundation for future studies on BGO crystal waveguides in electric field measurement applications.

## 1. Introduction

Electric field sensors find extensive application in static electric field detection and protection, relay protection, lightning disaster warning, electromagnetic compatibility, and various other fields [1,2,3,4,5]. Optical electric field sensors [6,7], owing to their compact size, light weight, high bandwidth, and robust resistance to interference from metal devices, have had increasing interest in recent years. These sensors operate on the principle of the Pockels effect [8,9,10,11,12], wherein the refractive index of the medium varies proportionally with the external electric field strength. Moreover, being constructed from insulating materials, optical electric field sensors minimize interference with electric field measurements [13,14].

Electro-optic materials constitute the core of optical electric field sensors, and their material characteristics profoundly influence the operational behavior of the entire device. Presently, numerous electro-optic materials are available for electric field sensors, with the most prominent ones being hard crystals like lithium niobate (LiNbO_3_, LN) [15,16,17], BGO, and bismuth silicon oxide (Bi_12_SiO_20_, BSO), with LN crystals being the most extensively researched [18]. The BGO material theoretically exhibits minimal natural birefringence (or significantly reduced through a processing technique [19]), optical rotation, pyroelectric effects, etc. Although BGO boasts a smaller electro-optic coefficient compared to LN, it can achieve a larger half-wave electric field, consequently elevating the upper limit of the electric field measurement. Despite the potentially decreased inherent electro-optic modulation sensitivity due to the smaller coefficient, sensitivity can be enhanced through judicious device and peripheral component designs while still maintaining high measurement upper limits. Additionally, BGO crystals exhibit excellent temperature stability, showing certain material advantages upon investigation.

In recent years, optical electric field sensors have predominantly relied on bulk crystal structures, employing single or double crystal configurations for electric field measurement. Quasi-reciprocal closed-loop voltage measurement schemes based on a Pockels effect box using a bismuth germanate crystal were proposed from 2001 to 2017 [20]. However, the majority of the studies focused on optimizing the demodulation scheme, with limited attention paid to structural optimization of the sensing crystal probe itself and targeted designs for spatial electric field measurements. To counteract environmental influences and inherent material defects, a Pockels effect box optical voltage sensor (OVS) with a dual-crystal structure probe was proposed in 2012 [21]. In 2020, Yansong Li et al. designed optical DC electric field sensors based on the Pockels effect using rotating BGO crystals [22]. Nevertheless, these solutions require two identical crystals or fiber optic rotary joints, which lack compactness and limit their further development. Although the BGO OVS based on a resonant cavity [22] can reduce the device size and enhance sensitivity, its accuracy is decreased. In addition to the bulk crystal approach, fiber-based OVS have been realized on bismuth germanate crystals from 2011 [23] to 2013 [24]. However, due to the high difficulty in preparation, its widespread adoption is currently challenging. Overall, sensors employing bulk crystal structures exhibit significant drawbacks, including complex sensor structures with large sizes, low measurement sensitivity, and limited frequency bandwidths.

With the advancement of integrated optics technology, electric field sensors based on integrated optics have garnered widespread attention, typically utilizing optical waveguides integrated on planar substrates. Optical waveguides are commonly fabricated on LN substrates with antennas or electrodes positioned near the waveguides through photolithography. By tailoring specific antennas and electrodes, optical field sensors with high sensitivities of hundreds of μV/m or even tens of μV/m and frequency bandwidths from the kHz to the GHz level can be achieved. In 2006, Zeng Rong et al. designed LN optical waveguide electric field sensors [25,26], and Fushen Chen et al. developed various straight-through and reflective LN asymmetric MZI sensors and systems in 2016 [27,28]. Furthermore, Fushen Chen et al. attempted a new scheme to control the operating point by utilizing the refractive index difference between two waveguide arms in 2020 [29], but it is difficult to accomplish the exact effective refractive index of the waveguide in accordance with the design value in actual processing. In 2020, Zhang Jiahong et al. developed an asymmetric MZI electric field sensor equipped with a tapered antenna array to achieve a frequency response of 100 kHz~26 GHz, but with significant response fluctuations of about ±10 dB [22]. Nevertheless, LN crystals, being of a triclinic crystal system, exhibit optical rotation and natural birefringence, resulting in reduced sensor sensitivity due to depolarization. In contrast, without natural birefringence and pyroelectric effects, BGO crystals offering superior sensing characteristics and mechanical stability. C. Miese et al. discussed the potential of this technique for the fabrication of advanced sensor arrays for voltage sensing applications [30]. Hence, the exploration of integrated optical electric field sensors based on BGO crystals holds significant promise. Furthermore, leveraging the M–Z interference principle, a Y-waveguide branch coupler was designed within the crystal, converting the phase difference measurement into a light intensity measurement, thereby eliminating unnecessary optical device coupling, enhancing sensor measurement accuracy, and reducing design costs. The primary focus of this article is to investigate the optical waveguide characteristics of BGO crystals for electric field sensors.

This article aims to comprehensively research and analyze the BGO crystal optical waveguide electric field sensor from the following perspectives: first, the electro-optic effect of the crystal is theoretically analyzed. Second, we carry out the design and analysis of the integrated crystal optical waveguide structure based on the M–Z interferometer principle. Finally, a femtosecond laser is utilized to conduct experiments on the inscription of waveguides on BGO crystals.

## 2. The Working Principle of an Integrated Optoelectronic Field Sensor

### 2.1. Electro-Optic Effect of BGO Crystal

Given the isotropic nature of BGO crystals, the second-order equation describing the refractive index ellipsoid in the absence of an external electric field can be expressed as
(1)$${\beta_{1}}^{2}X_{1}^{2}+{\beta_{2}}^{2}X_{2}^{2}+{\beta_{3}}^{2}X_{3}^{2}=1$$

Equation (1) can be written in the principal axis coordinate system of the refractive index ellipsoid as shown in Equation (2).
(2)$$\frac{{X_{1}}^{2}}{{n_{1}}^{2}}+\frac{{X_{2}}^{2}}{{n_{2}}^{2}}+\frac{{X_{3}}^{2}}{{n_{3}}^{2}}=1$$

When the light wave vector is $\vec{k}$, a plane perpendicular to passing through the origin of the coordinate system intersects the ellipsoid, and the semi-lengths of the short and long axes of the resulting ellipse represent the refractive indices of the two eigenwaves of the light wave vector $\vec{k}$. The directions of the short and long axes are the polarization directions of the two eigenwaves.

In the natural state without an external electric field, the refractive index ellipsoid of an isotropic BGO crystal is a sphere, with the principal axis coinciding with the crystal axis. Equation (1) becomes $\beta_{1}=\beta_{2}=\beta_{3}=B^{0}=\frac{1}{{n_{0}}^{2}}$, where $B^{0}\;$is the inverse dielectric tensor of the BGO crystal in its natural state and $n_{0}$ is the refractive index of the crystal.

The electro-optic coefficient of BGO crystals has only one independent component, $\gamma_{41}$ [30]. When an electric field *E* is applied to the BGO crystal, the electro-optic effect matrix can be expressed as shown in Equation (3), where $B_{i}$ represents the inverse dielectric tensor after applying the electric field and $E_{i}$ represents the component of the electric field along the principal axis $X_{i}$.
(3)$$\left[\begin{array}{c} \Delta B_{1}\\ \begin{array}{c} \Delta B_{2}\\ \begin{array}{c} \begin{array}{c} \Delta B_{3}\\ \Delta B_{4} \end{array}\\ \Delta B_{5} \end{array} \end{array}\\ \Delta B_{6} \end{array}\right]=\left[\begin{array}{c} B_{1}-{B^{0}}_{1}\\ \begin{array}{c} B_{2}-{B^{0}}_{2}\\ \begin{array}{c} \begin{array}{c} B_{3}-{B^{0}}_{3}\\ B_{4} \end{array}\\ B_{5} \end{array} \end{array}\\ B_{6} \end{array}\right]=\left[\begin{array}{c} 0\\ \begin{array}{c} 0\\ \begin{array}{c} \begin{array}{c} 0\\ \gamma_{41} \end{array}\\ 0 \end{array} \end{array}\\ 0 \end{array}\begin{array}{c} 0\\ \begin{array}{c} 0\\ \begin{array}{c} \begin{array}{c} 0\\ 0 \end{array}\\ \gamma_{41} \end{array} \end{array}\\ 0 \end{array}\begin{array}{c} 0\\ \begin{array}{c} 0\\ \begin{array}{c} \begin{array}{c} 0\\ 0 \end{array}\\ 0 \end{array} \end{array}\\ \gamma_{41} \end{array}\right]\left[\begin{array}{c} E_{1}\\ E_{2}\\ E_{3} \end{array}\right]$$

Therefore, the refractive index ellipsoid will change to Equation (4) under the influence of an external electric field.
(4)$$\frac{{X_{1}}^{2}}{{n_{0}}^{2}}+\frac{{X_{2}}^{2}}{{n_{0}}^{2}}+\frac{{X_{3}}^{2}}{{n_{0}}^{2}}+2\gamma_{41}\left(E_{1}X_{2}X_{3}+E_{2}X_{3}X_{1}+E_{3}X_{1}X_{2}\right)=1$$

The equation above contains cross-terms, indicating that the coordinate system of the refractive index ellipsoid no longer coincides with the one before applying the electric field. When the *E* direction is parallel to the <110> crystal orientation, $E_{1}=E_{2}=E/\sqrt{2}$, $E_{3}=0$, and Equation (4) becomes:(5)$$\frac{{X_{1}}^{2}}{{n_{0}}^{2}}+\frac{{X_{2}}^{2}}{{n_{0}}^{2}}+\frac{{X_{3}}^{2}}{{n_{0}}^{2}}+\sqrt{2}\gamma_{41}E\left(X_{2}X_{3}+X_{3}X_{1}\right)=1$$

The equation above indicates that under the influence of an external electric field, the principal axes of the BGO crystal’s refractive index ellipsoid rotate. The new principal axes of the refractive index are represented as ${X_{i}}^{\prime }$. By performing a coordinate transformation on Equation (5), the coefficients of the refractive index ellipsoid become:(6)$$\left\{\begin{array}{l} \Delta B_{1}={B_{1}}^{\prime }\\ \Delta B_{2}={B_{2}}^{\prime }--{B^{0}}_{1}=\gamma_{41}E{B^{0}}_{1}=-\gamma_{41}E\\ \Delta B_{3}={B_{3}}^{\prime }-{B^{0}}_{1}=0 \end{array}\right.$$

Therefore, after applying an electric field along the <110> direction, the new principal refractive indices of the BGO crystal are:(7)$$\left\{\begin{array}{l} {n_{1}}^{\prime }=n_{0}-\frac{1}{2}{n_{0}}^{3}\gamma_{41}E\\ {n_{2}}^{\prime }=n_{0}+\frac{1}{2}{n_{0}}^{3}\gamma_{41}E\\ {n_{3}}^{\prime }=n_{0} \end{array}\right.$$

The rotation angle of the refractive index ellipsoid can be determined by solving for the direction of the principal axes, and the relationship between the semiaxes’ lengths of the new principal axes under the applied electric field and the original principal axes is given by:(8)$$\left[\begin{array}{ccc} 1/2 & 1/2 & 1/\sqrt{2}\\ 1/2 & 1/2 & -1/\sqrt{2}\\ 1/\sqrt{2} & -1/\sqrt{2} & 0 \end{array}\right]$$

From the above results, it can be seen that the rotation angle of the principal axes of the refractive index ellipsoid is independent of the magnitude of the electric field but depends on its polarity. Under the electric field along the <110> direction, the BGO crystal becomes a biaxial structure, and the two optical axes lie on the plane determined by the new principal axes ${X_{1}}^{\prime }$ and ${X_{2}}^{\prime }$. The positional relationship between the new coordinate system of the principal axes of the refractive index ellipsoid and the original coordinate system is shown in Figure 1.

From Equation (7) and Figure 1, it can be seen that when an electric field is applied along the <110> direction, the light propagating along ${X_{3}}^{\prime }$ will obtain refractive indices of ${n_{1}}^{\prime }$ and ${n_{2}}^{\prime }$, respectively. If the polarization direction of the incident light is along the <110> direction, the eigenwaves can be equally distributed to the axis ${X_{1}}^{\prime }$ and axis ${{\;X}_{2}}^{\prime }$. The polarization direction can also be only along ${X_{1}}^{\prime }$ or ${X_{2}}^{\prime }$, obtaining refractive indices of ${n_{1}}^{\prime }$ or ${n_{2}}^{\prime }$.

For two coherent light beams *A* and *B* that satisfy the interference condition, if their refractive indices are $n_{A}$ and $n_{B}$, respectively, and their optical paths are $l_{A}$ and $l_{B}$, respectively, the phase difference between them can be expressed as shown in Equation (9).
(9)$$\Delta \theta_{AB}=\frac{2\pi }{\lambda }n_{A}l_{A}-\frac{2\pi }{\lambda }n_{B}l_{B}$$

The change in refractive index caused by the Pockels effect of the BGO crystal under the influence of an external electric field can obtain the Pockels phase difference, which will cause a change in the intensity of the interference light.

### 2.2. Principle and Design of Integrated Optical Electric Field Sensor

The Mach–Zehnder interferometer (MZI) plays a significant role in integrated optical electric field sensors. As shown in Figure 2, the overall system of the integrated optical electric field sensor involves a polarized light emitted by a light source with its polarization direction parallel to the refractive index principal axis ${X_{1}}^{\prime }$ (or ${X_{2}}^{\prime }$) of the BGO crystal. The incident light traverses the crystal probe via the peripheral optical path, where it undergoes modulation by the measured electric field induced by a miniature antenna, leading to a Pockels phase difference. Subsequently, the Pockels phase difference, encoding the electric field information, is externally expressed as optical intensity information through interference. This optical signal is further transmitted via an optical fiber along the peripheral optical path and directed into an optoelectronic conversion module for transformation into an electrical signal. Following demodulation processing, the electric field measurement is accomplished.

Traditional Mach–Zehnder interferometers (MZIs) achieve the splitting and coupling of coherent light via two Y-branch couplers, constituting a dual-beam interferometer. Moreover, researchers, including Ichikawa, have explored reflective MZIs for electric field detection [31]. This variant of MZI, depicted in Figure 3, offers advantages such as expanded measurement range, reduced crystal size, and optimized device structure. The laser emits coherent light, evenly divided by the two Y-branch couplers, which is simultaneously directed to the signal arm and the reference arm. Upon the application of an external electric field to the signal arm via electrodes flanking it, a phase delay occurs during the measurement process. In our design, the second Y-branch coupler is substituted with an end-reflecting film, where light is reflected and redirected along its original trajectory to Port 0. Within this configuration, the electric fields at Port 3 and Port 4 can be expressed as
(10)$$E_{3}=\frac{\sqrt{2}}{2}A_{0}exp\left(j\left(\omega t-kn_{0}l_{1}+\varepsilon_{1}+\Delta \varphi \right)\right)$$
(11)$$E_{4}=\frac{\sqrt{2}}{2}A_{0}exp\left(j\left(\omega t-kn_{0}l_{2}+\varepsilon_{2}\right)\right)$$
where $A_{0}$ is the amplitude of the incident light, ω is the frequency of the incident light, k is the propagation constant, $n_{0}$ is the refractive index of the optical waveguide, $l_{1}$ and $l_{2}$ are the lengths of the two waveguide arms, and $\varepsilon_{1}$ and $\varepsilon_{2}$ are the phase errors caused by the fabrication process or operating point drift of the two waveguide arms. Δφ is the phase introduced to the ends of the electrodes due to the applied voltage.

After the interference of the two beam Y-branch couplers, the output intensity can be represented as
(12)$$I_{out}=\frac{1}{2}I_{0}\left(1+\cos\left(\varphi_{0}-\Delta \varepsilon -2\Delta \varphi \right)\right)$$
where $\varphi_{0}=kn_{0}l_{2}-kn_{0}l_{1}$ and $\Delta \varepsilon =\varepsilon_{2}-\varepsilon_{1}$. Due to the reflection film causing the incident light to transmit twice in the electric field region, the Pockels effect of the BGO crystal introduces a phase delay of 2Δφ.

The output optical intensity of the electric field sensor can be demodulated to determine the strength of the applied electric field. The fabrication of the optical waveguide is a crucial step in realizing the electric field sensor. In order to prepare a well-guiding Y-branch optical waveguide, this experiment focuses on the fabrication of waveguides to obtain optimal fabrication parameters. These parameters are then transferred to the Y-branch waveguide, significantly reducing the complexity of this experiment. Therefore, the following experiment investigated the femtosecond laser writing of waveguides.

## 3. Optical Waveguide Writing Experiment

### 3.1. Experiment Platform

Although cladding waveguides based on femtosecond laser writing technology have been previously achieved on BGO crystals, the diameter of the inscribed waveguides was 100 μm, resulting in multimode transmission [32], contrasting markedly with the 8 μm cladding waveguides prepared in this study. As size diminishes, however, the interaction between cladding traces increases the susceptibility to waveguide structure damage [33]. Consequently, three continuous writing experiments, among which Experiment I and Experiment II employed the same processing system A and Experiment III utilized an alternative processing system B (comprising femtosecond laser writing and in situ refractive index characterization), were conducted to ascertain suitable waveguide writing parameters for this study.

The structure of the femtosecond laser inscription experimental platform for the three experiments is illustrated in Figure 4. For Experiments I and II, a Spectra-Physics® (Innofocus, Heidelberg, Australia) laser served as the light source, operating at an output center wavelength of 520 nm and a pulse width of 300 fs. The BGO crystals used in these experiments had dimensions of 10 mm × 12 mm × 2 mm, with a refractive index of approximately 2.04 at 1550 nm. A microscope objective with 40×magnification and a numerical aperture of NA 0.65 were employed. Both experiments utilized transverse inscription, resulting in a single inscription trace width of 3 μm. Depending on the inscription parameters, the depth of a single trace varied from 10 to 50 μm. The number of cladding strips and their arrangement are depicted in Figure 5. Experiment III employed a NanoLAB Holoview 3D-Ri (H3D) femtosecond laser processing system from Innofocus^®^ Australia (Heidelberg), featuring an output laser center wavelength of 515 nm and a pulse width of 330 fs. The BGO crystals used in this experiment measured 10 mm × 12 mm × 0.4 mm. Transverse inscription was performed using a microscope with 50× magnification and a numerical aperture of NA 0.9.

In Experiment I, several groups of suitable parameters were selected from reference [32]. To investigate the impact of repetition frequency on waveguide morphology, in Experiment II, the crystal was inscribed at a repetition frequency of 200 kHz, while Experiment I used a repetition frequency of 100 kHz as a control. Notably, the processing system in Experiment III was equipped with an in situ refractive index characterization platform, which enabled the direct acquisition of refractive indices of traces during the inscription process, thereby facilitating more efficient selection of appropriate inscribing parameters.

### 3.2. Writing Experiments at 1 kHz~200 kHz Repeat Frequency

In Experiment I, the crystal was inscribed using repetition frequencies of 1 kHz, 10 kHz, and 100 kHz and scanning speeds of 0.5 mm/s and 2 mm/s, and the cladding waveguide morphology end views with different inscribing parameters are shown in Table 1 and Table 2, where the waveguide depths of the top row in the figure were 60 μm, and the waveguide depths of the bottom row were 110 μm.

Taking 100 kHz as a control, in Experiment II, the crystal was inscribed using 200 kHz repetition frequency, with scanning speeds of 0.1 mm/s, 2 mm/s, and 3 mm/s, and waveguide depths of about 100 μm, and some of the waveguide end face topographic views obtained are summarized in Table 3, which shows that the crystal end face has been fine-polished and is no longer curved compared to the end-face view in Table 1.

In Experiment II, several sets of bilinear and index-elevated waveguides were also inscribed using the inscribing parameters of 1 kHz repetition frequency, 0.5 mm/s scanning speed, 150 nJ pulse energy, and trace spacing, and their end face and top view morphologies are shown in Table 4. It can be seen that compared with the cladding-type waveguide, the refractive-index-elevated waveguide core layer traces shown in Table 1 obviously present a more translucent state, indicating that the refractive index change of such traces is more likely to be positive.

As can be seen from Table 1 and Table 2, for the same waveguide, the longitudinal displacement between different inscribed traces is small, and the traces do not have much difference; when the difference in waveguide depth occurs, the inscribed results under the same parameter change drastically, and especially at the lower repetition frequencies of 1 kHz and 10 kHz, the effect of waveguide depth is even greater; when the waveguide is located at the position of about 60 μm depth, the longitudinal depth of the traces increases even at the same pulse energy. Table 1 and Table 4 show the waveguide end face at 100 kHz and 2 mm/s inscription parameters, and there is not much difference between the inscription results of waveguide depths of 110 μm and 100 μm at approximate pulse energies. Therefore, it is easier to form a good waveguide when the waveguide depth is controlled to around 100~110 μm.

With other parameters kept constant, by observing the end face morphology diagrams in Table 1 and Table 3, it can be found that the cladding waveguide morphology is gradually clear and moderately wrapped when using the inscription parameters of 10 kHz and 0.5 mm/s, as well as those of 100 kHz and 200 kHz; however, as the pulse energy is increased, the width and longitudinal depth of the traces are enlarged, so that the waveguide core layer starts to blur and loses its light-guiding ability; at a repetition frequency of 1 kHz, even if the pulse energy is increased to 3.1 μJ, the average power is still small, resulting in poor waveguide morphology. Therefore, in order to obtain a good waveguide structure, it is necessary to use the appropriate pulse energy for different repetition frequencies. In the light-guiding tests on waveguides, we found that the cladding-type waveguides inscribed with 100 kHz repetition frequency and 41 nJ~96 nJ pulse energy, and 200 kHz repetition frequency and 45 nJ~74 nJ pulse energy have good morphology and light-guiding performance.

The topographic integrity of the waveguide and light-guiding performance are also affected by the scanning speed. When the scanning light speed is slow, the stress area increases, so a scanning speed of 50 μm/s is generally used for BGO bilinear waveguides. In order to inscribe the cladding-type waveguide where the core layer is not affected by stress as much as possible, the scanning speed should be increased. Table 1 and Table 3 illustrate that, when other inscribing parameters are unchanged, increasing the scanning speed will weaken the modification ability of the laser pulse on the material, and the inscribed traces will become more sparse, while at low repetition frequency, the scanning rate should be reduced to obtain uniform and clear trace distribution. Comparing the two scanning speeds of 0.5 mm/s in Table 1 and 0.1 mm/s in Table 3, it is found that even though the trailing defects in the traces cover the core layer region in Table 3, the light-guiding performance is still normal, and it is hypothesized that the trailing defects may have formed a rather low refractive index, which increases the difference in refractive indices between the remaining core layer and the traces and makes it possible to guide light in the smaller core layer space.

After further examination, it was finally obtained that at 100 μm waveguide depth and 0.1 mm/s scanning speed, the cladding-type waveguide inscribed with two groups of parameters of (1) repetition frequency of 100 kHz and average power of 7.42 mW (pulse energy of 74.2 nJ) and (2) repetition frequency of 200 kHz and average power of 10.15 mW (pulse energy of 50.8 nJ) had good light-guiding performance. However, due to the higher pulse energy used in the former group, the traces were trailing, covering part of the core layer area and changing the core layer material properties, so the optimal parameters for this experiment are determined to be the second group. After the waveguide loss test, the loss per unit length of the waveguide inscribed under the second group of parameters was about 1.2 dB/cm. In future experiments, if a higher repetition frequency is used, the pulse energy can be appropriately lowered to avoid the trailing of the longitudinal stretching depth of the trace to destroy the core layer structure, which can further reduce the waveguide loss.

### 3.3. In Situ Refractive Index Characterization Experiment

Refractive index reconstruction is a computational method for deriving the refractive index distribution from optical measurements, which calculates the refractive index distribution by measuring the intensity of the transmitted beam near the end face, obtaining the refractive index change in the femtosecond laser exposure track. Then, a CCD camera is used to photograph the near-field light intensity distribution at the end face of the optical waveguide to determine the normalized electric field component of that intensity.

In this paper, the mode-field distribution reconstruction (shown in Figure 6) of the light intensity distribution map (shown in Figure 7) of a cladding-type waveguide was performed, and it was determined that the refractive index change of the cladding-type waveguide inscribed traces was about −0.0016, which is within the design range.

In this paper, the refractive index of five inscribed traces with different average power and scanning speeds were also directly characterized, and the results are shown in Figure 8, where the color bar indicates the amount of change in refractive index, which is positive and negative, and the darker the color is, the more obvious the negative change in refra-tive index is.

Figure 8a shows that a certain degree of negative refractive index change occurs at and near the waveguide inscription, and the reduction is about 0.001~0.008, which is proportional to the average inscription power. Further observation of Figure 8a,b reveals that the refractive index reduction area is surrounded by a ring of refractive index increase area at the same time, which may be caused by the lattice extrusion that produces a stress area, and when the scanning speed decreases, the range of this refractive index increase area expands. Although the stress-induced refractive index elevation seems to be unavoidable, the use of a larger pulse energy and faster scanning speed can make the size of the stress zone smaller than the size of the core layer of the waveguide, and the effect of the stress zone on the core layer can be avoided by superimposing multiple traces, which will cover the low refractive index region from the high refractive index region.

## 4. Design and Fabrication of Antennas and Electrodes

### 4.1. Analysis of Structural Parameters of Antennas and Electrodes

Currently, the micro-antennas of optical electric field sensors mainly consist of segmented electrode antennas, conical dipole antennas, and bowtie antennas. Among them, the structure of the conical dipole antenna (CDA), a type of traveling wave antenna which is generally used to avoid the oscillation of the antenna surface current between the end and the feed point, enabling the sensor to achieve a larger bandwidth, changes from the bottom to the top in a conical shape. The above structure is equivalent to loading different impedances at different positions, resulting in a traveling wave distribution of the antenna surface current [32]. Since the polarization direction of a CDA is parallel to the height direction of the antenna, if the line connecting the tip and the midpoint of the base is designed to be at a certain angle relative to the base, the inclined conical dipole antenna can obtain a new polarization direction at this angle. Therefore, the design of a CDA is more flexible, and its volume is smaller compared to other electrode forms, which is convenient for combination, placement, and expanded design.

The plane structure of a CDA is shown in Figure 9, where the antenna height is $H_{a}$, the antenna width is $W_{a}$, the electrode width is $W_{el}$, and the electrode spacing is$\;G_{el}$. All antenna models simulated in this section were constructed on a BGO crystal substrate and included an air domain in the computational domain. The side view of the waveguide made of BGO and electrode combination made of gold is shown in Figure 10.

### 4.2. Antenna Electrode Process Scheme

The antenna designed in this article is primarily intended to be paired with waveguides inscribed within a BGO crystal using femtosecond laser writing. Conventional surface electrode deposition techniques as shown in Figure 11 can meet the requirements. Prior to electrode fabrication, the upper surface of the crystal is thinned and removed, and a high-power femtosecond laser pulse is used to ablate an air groove with a depth of approximately 10 μm to serve as the deposition groove for the embedded electrode [33]. Common processes for depositing thin metal layers include magnetron sputtering, electron beam evaporation, and electrochemical deposition. However, for the electrodes designed in this article with a total thickness of 12 μm, the efficiency of conventional sputtering or evaporation methods is relatively low. Therefore, after forming the antenna electrode pattern on the surface of the BGO substrate using photolithography, the antenna electrode was deposited by electrochemical deposition. Gold (Au) was selected as the deposition material, and any excess metal deposited along with the photoresist was removed by lift-off due to its resistance to corrosion. Considering the weak adsorption force between gold molecules and the substrate, a thin layer of chromium (Cr) with a thickness of approximately 400 Å was evaporated prior to the deposition of gold to enhance the adhesion of the antenna electrodes to the substrate. The electrode fabrication process is illustrated in Figure 11. Previous LN sensors typically required the deposition of a 50–800 nm thick SiO_2_ layer on the upper surface of the waveguide to isolate it from the surface antenna electrode overlaying it. However, in this article, since the cladding-type waveguide of the BGO still maintains a distance of at least 1 μm from the upper surface after thinning, there was no need for the deposition of an SiO_2_ buffer layer [34]. This simplifies the process flow while also reducing stress caused by temperature-induced differences in material expansion.

### 4.3. Electro-Optic Effect Measurement

The performance of the developed device as a fully integrated tunable electro-optic modulator was examined by applying a DC voltage in the range of 0 to 20 V to the microelectrodes. The electrodes were connected to a power supply through microelectrode contacts made of 10 μm thick gold wires, and the electrode lead bonding was achieved using a WESTEBOND microelectrode device. The welding quality was ensured by checking the connectivity of the electrode structure and microscope analysis. The laser beam was polarized at an angle of 45° relative to the crystal axis with the help of a half-wave plate. In this configuration, the beam propagates along the optical axis. As shown in Figure 12, the electrodes were positioned so that the electric field was transverse to the beam propagation direction. Without an external voltage, the two polarization components of the incident light propagate through the waveguide at the same speed. However, when an external voltage is applied, a phase shift is introduced between the components. The beam at the output of the waveguide was analyzed by a Glan Thompson polarizer placed crosswise to the input polarizer. Bonding quality was assured by checking connectivity and microscopic analysis of electrode structures as shown in the figure.

The gain of a single specific antenna electrode structure for the electric field to be measured is ξ, and the electric field with a half wavelength between electrodes is ξ times that of the spatial electric field with a half wavelength (non-array case) $E_{\pi }$. Calculations based on antenna impedance, electro-optical overlap factor, and equivalent circuit transfer function indicate that it is related to antenna width $W_{a}$, antenna height $H_{a}$, electrode length $L_{el}$, electrode spacing $G_{el}$, and electrode width $W_{el}$, and it is affected by these parameters to varying degrees. Given the interconnectedness of various indicators for electric field sensors, a general analysis of the impact of antenna electrode parameters on is now presented.

Using a 1 GHz, 1 V/m background electric field in the z-direction as excitation, the position of the point probe was set to the midpoint of the modulation region in the direction of light transmission (−D−5 μm, 0, 0). The electric field gain value at the probe position, ξ, is equal to the electric field strength at that point E. We first analyzed the antenna parameters$\;W_{a}$ and $H_{a}$. When fixed at D=3 μm, $H_{a}=2\;mm$, $L_{el}=1\;mm$, $G_{el}=12\;\mu m,{and\;W}_{el}=5\;\mu m$, we obtained varying yields for the gain and half-wavelength electric field as a function of antenna width, as shown in Figure 13a. Keeping other parameters constant and varying $W_{a}=100\;\mu m$ or 200 μm, the gain and half-wavelength electric field as a function of antenna height are plotted in Figure 13b.

As shown in Figure 13a, increasing the antenna width $W_{a}$ from 50 μm to 650 μm enhances the electric field gain, resulting in a relatively small decrease in the half-wavelength electric field $E_{\pi }$ from 4377.9 kV/m to 4025.4 kV/m. Figure 13b demonstrates that when $H_{a}$ is small, $E_{\pi }$ is highly sensitive to changes in antenna height. When $W_{a}$ is set to 100 μm and 200 μm, respectively, increasing $H_{a}$ from 0.5 mm to 1 mm alone results in a decrease of 19,550 kV/m and 16,379 kV/m in $E_{\pi }$, corresponding to a reduction rate of 63% and 59%, respectively. However, when $H_{a}$ exceeds 1.5 mm, the rate of decrease in $E_{\pi }\;$ gradually slows down as $H_{a}$ increases. The small differences between the curves for different values in Figure 13b further support the conclusion from Figure 13a that the antenna width has a relatively minor impact on $E_{\pi }$. This suggests that a suitable reduction in antenna height during design can lead to a larger half-wavelength electric field. Additionally, considering the interference of metal components on the measured electric field, reducing the antenna width can minimize the size of the antenna without significantly affecting the half-wavelength electric field or sensitivity, ensuring a broad measurement range [35,36,37].

Next, we analyzed the electrode parameters $L_{el}$ and $G_{el}$. Setting D=3 μm, $W_{a}=200\;\mu m$, $G_{el}=12\;\mu m$, $W_{el}=5\;\mu m$, and setting $H_{a}\;$to 3 μm and 200 μm, and varying $H_{a}\;$between 1 mm and 3 mm, Figure 14 illustrates the changes in gain and half-wavelength electric field as a function of electrode length. Keeping other parameters constant and varying $G_{el}$, Figure 15a demonstrates the variation of the half-wavelength electric field with electrode spacing. Similarly, Figure 15b shows the changes in the half-wavelength electric field as electrode width is altered [38,39]. Figure 14 reveals that increasing $L_{el}\;$does not lead to a greater electric field gain, but rather results in poorer antenna electrode response due to impedance mismatch. However, since the half-wavelength electric field and sensitivity are ultimately determined by the gain-length product, increasing $L_{el}$ within the simulation range enhances $\xi \times L_{el}$ and overall modulation efficiency, leading to a corresponding decrease in $E_{\pi }$. The figure also indicates that the impact of changes in $L_{el}\;$on $E_{\pi }$ is greater when $H_{a}$ is smaller. When $H_{a}=1\;mm$, increasing $L_{el}$ from 1 mm to 10 mm results in a decrease in $E_{\pi }$ from 11,410 kV/m to 6339 kV/m, representing a reduction of approximately 44%. Conversely, when $H_{a}=3\;mm$, the decrease in $L_{el}$ from 2332 kV/m to 1601 kV/m upon increasing $E_{\pi }$ from 1 mm to 10 mm is approximately 31%.

In Figure 15, an increase in electrode spacing also results in a significant increase in the half-wavelength electric field. As the electrode spacing increases from 12 μm to 22 μm and 62 μm, the half-wavelength electric field increases by approximately 51% and 196%, respectively. However, changes in electrode width have a minimal impact on the half-wavelength electric field. Under different values of $H_{a}$, the variation in $E_{\pi }$ caused by changes in $W_{el}$ does not exceed 16%. Based on the above results, it is evident that simply increasing the electrode length does not significantly improve modulation efficiency, and it is more prone to impedance mismatch issues. Therefore, to enhance the sensitivity of the electric field sensor, a better approach is to utilize antenna arrays to increase the length of the modulation region. On the other hand, to obtain a larger half-wavelength electric field and thereby increase the measurement upper limit, increasing the electrode spacing can be effective, while appropriately increasing the electrode length can also be considered to balance sensitivity.

## 5. Conclusions

This article focuses on the research of optical electric field sensors using bismuth germanate (BGO) crystals as sensing probes. This research encompasses waveguide analysis and design, femtosecond laser waveguide fabrication, integrated sensor processing, and electro-optic effect measurement. By conducting three different inscription experiments using two femtosecond laser inscription platforms, this paper compares the refractive index changes and inscribed morphology of the waveguides under various inscription repetition rates, inscription depths, pulse energies, and scanning speeds, thereby obtaining the optimal inscription parameters for BGO optical waveguides applicable to electric field sensors. The small antenna electrode of the sensor is analyzed and designed, and its performance metrics are evaluated through electric field sensing experiments. The advantage of this waveguide sensor lies in the fact that it is the first time BGO crystal Y-waveguides have been utilized in an electric field measurement system, demonstrating its simplicity and reliability. Additionally, the refractive index is estimated using in situ refractive index characterization technology, showcasing the effectiveness of crystal waveguide inscription in optimizing measurement results and satisfying different requirements for precise electric field measurements. In the future, the system is expected to achieve integration, as bulk BGO crystals can potentially be replaced by BGO crystal Y-waveguides to facilitate on-chip multi-functional integration. Comprehensive experimental results indicate that the system is capable of measuring electric fields within a broad dynamic range. The designed electric field sensor boasts a response bandwidth of 9.4 GHz, with a measurement range of 2.5 mV/m to 39.3 mV/m for high-frequency fields and 42 mV/m to 131 kV/m for relatively low-frequency fields. This integrated, reconfigurable, and robust instantaneous electric field measurement system based on integrated BGO optical waveguide electric field sensors will meet the demands of future array electric field detection.

## Figures and Tables

**Figure 1 sensors-24-05570-f001:**
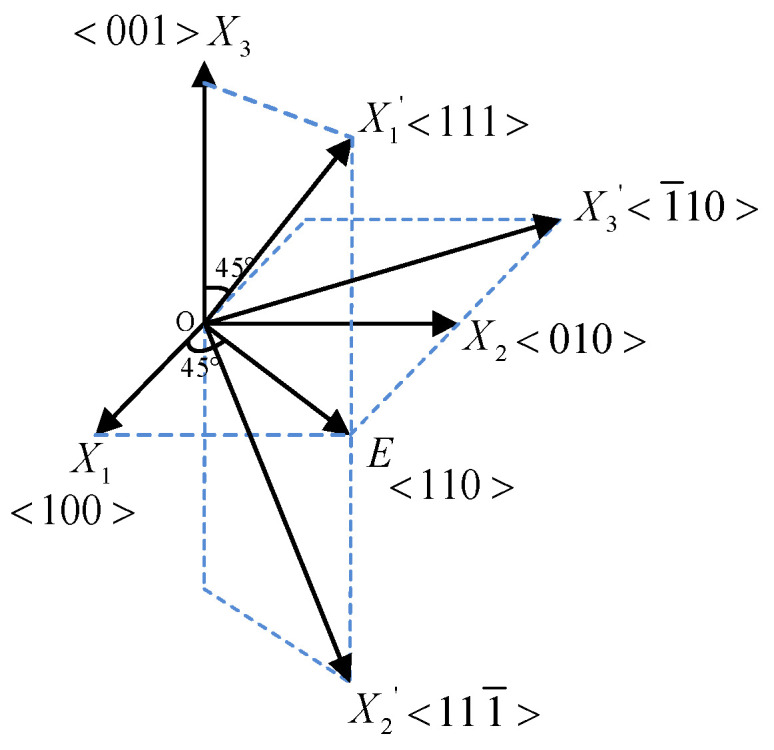
Schematic diagram of the change in refractive index ellipsoid of a BGO crystal.

**Figure 2 sensors-24-05570-f002:**
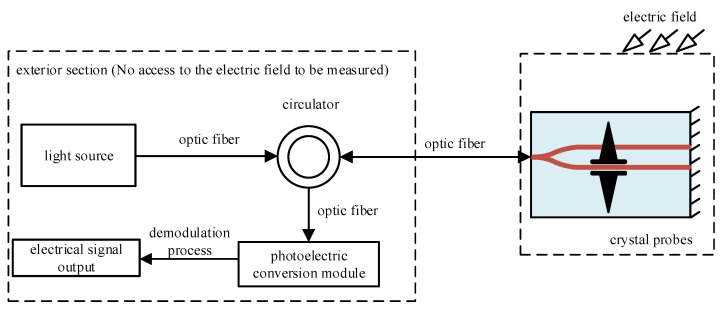
Integrated optical field sensor system.

**Figure 3 sensors-24-05570-f003:**
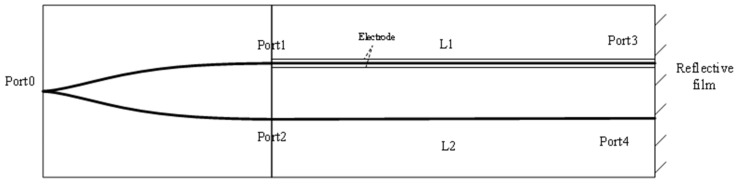
Reflective MZI structure.

**Figure 4 sensors-24-05570-f004:**
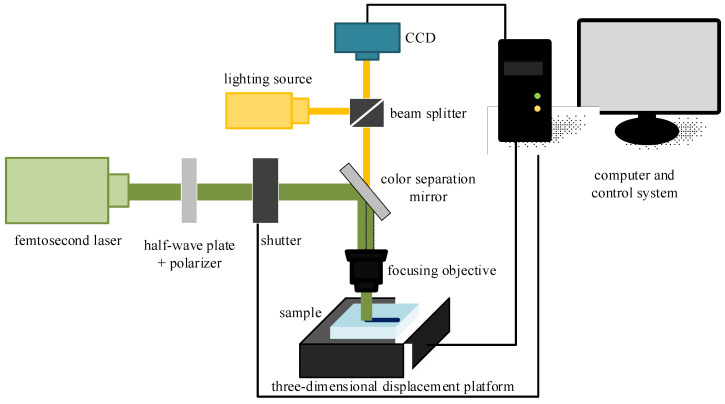
Schematic diagram of the experimental platform for femtosecond laser inscription.

**Figure 5 sensors-24-05570-f005:**
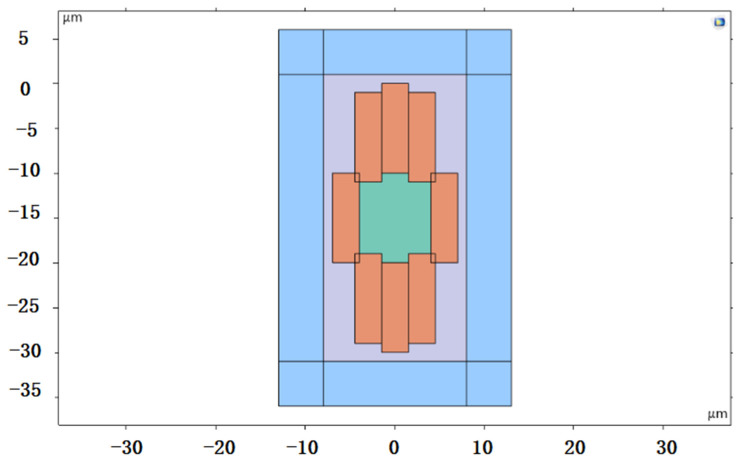
Simulation of actual waveguide end face topography.

**Figure 6 sensors-24-05570-f006:**
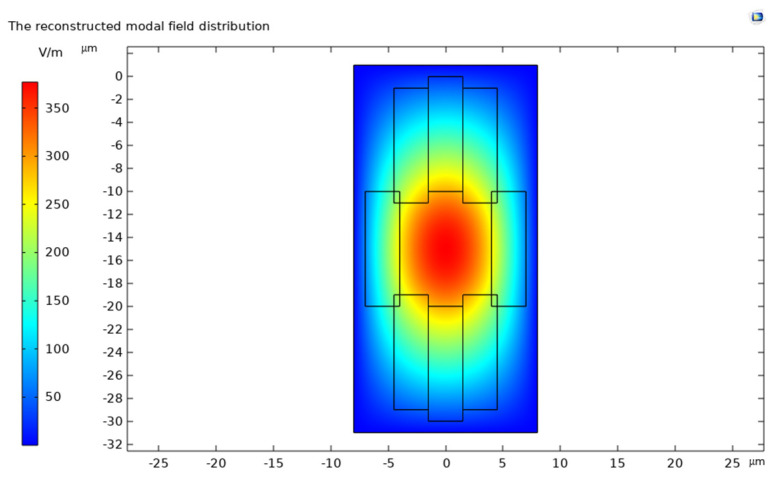
The reconstructed modal field distribution.

**Figure 7 sensors-24-05570-f007:**
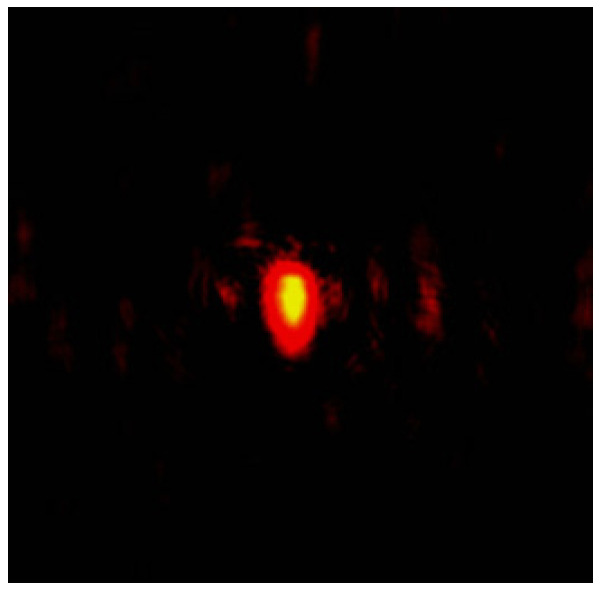
The near-field intensity distribution of the cladding-type waveguide.

**Figure 8 sensors-24-05570-f008:**
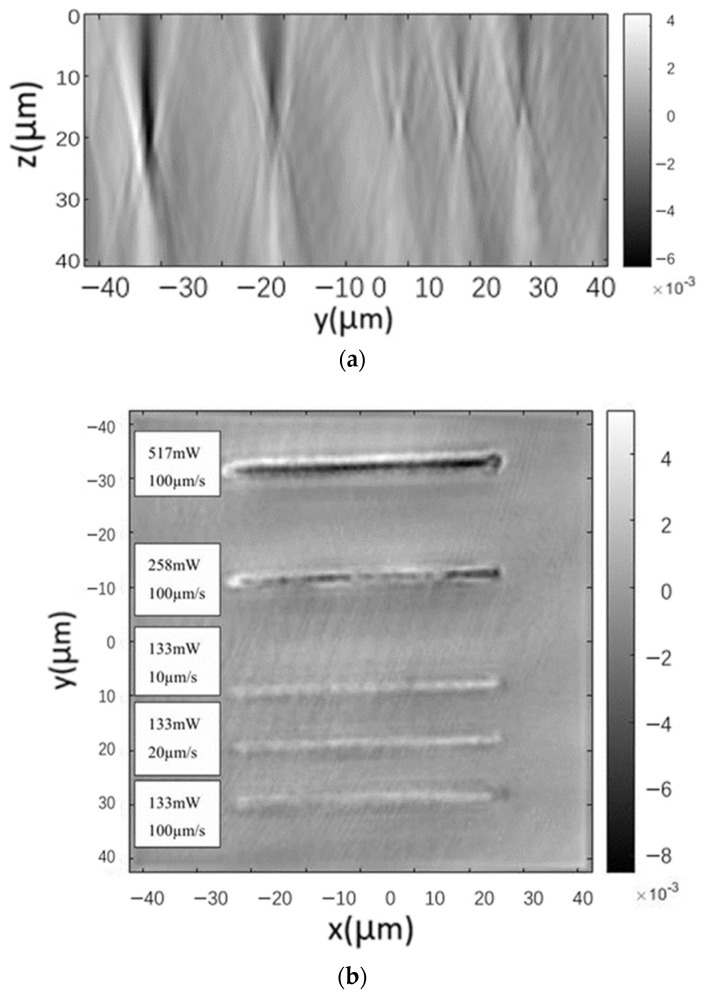
Results of refractive index characterization of the writing traces. (**a**) End face and (**b**) top-down view.

**Figure 9 sensors-24-05570-f009:**
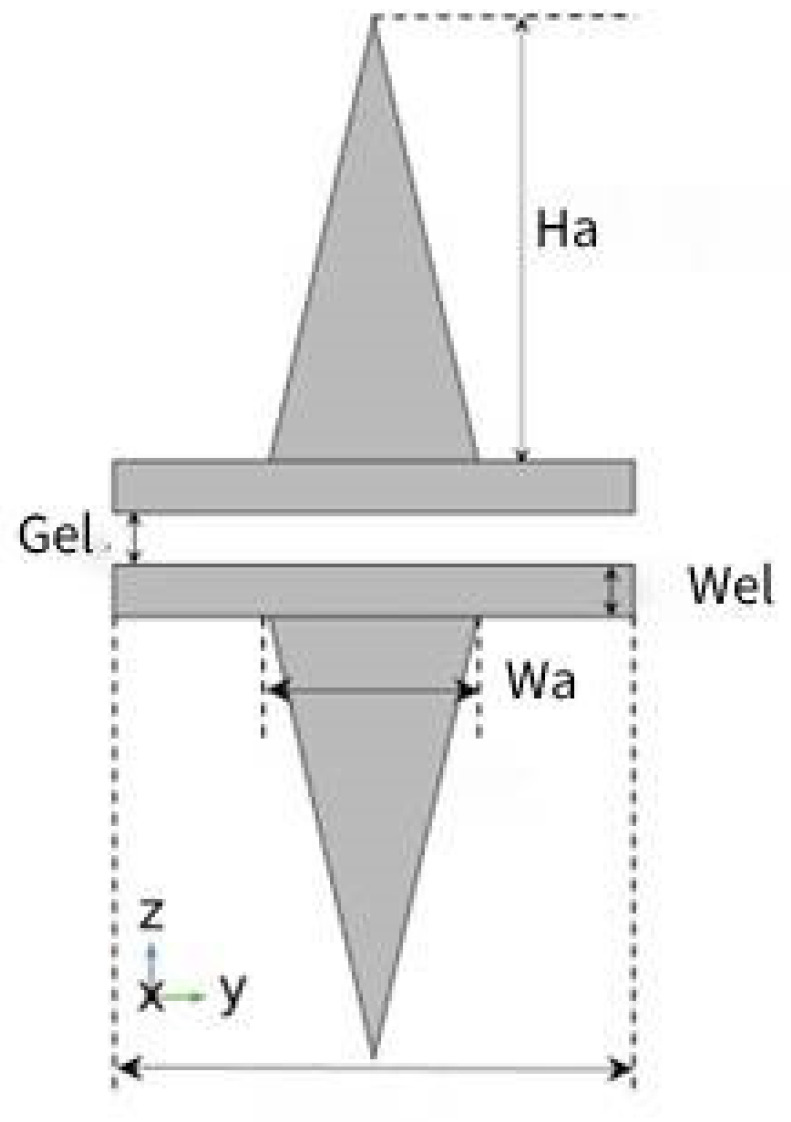
The structure of the conical dipole antenna.

**Figure 10 sensors-24-05570-f010:**
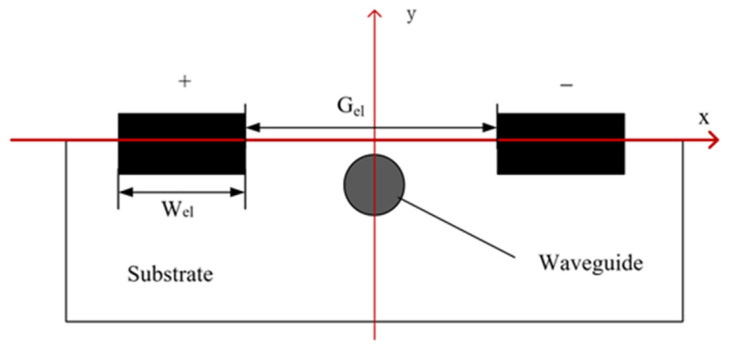
Side view of waveguide and electrode combination.

**Figure 11 sensors-24-05570-f011:**
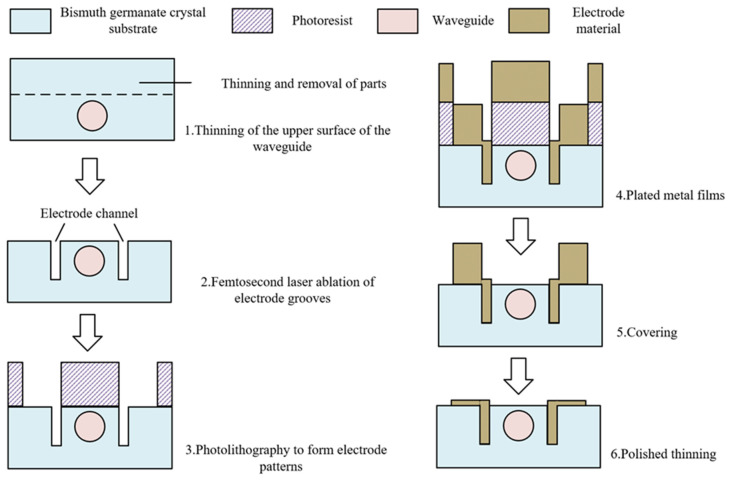
Antenna electrode process program flow.

**Figure 12 sensors-24-05570-f012:**
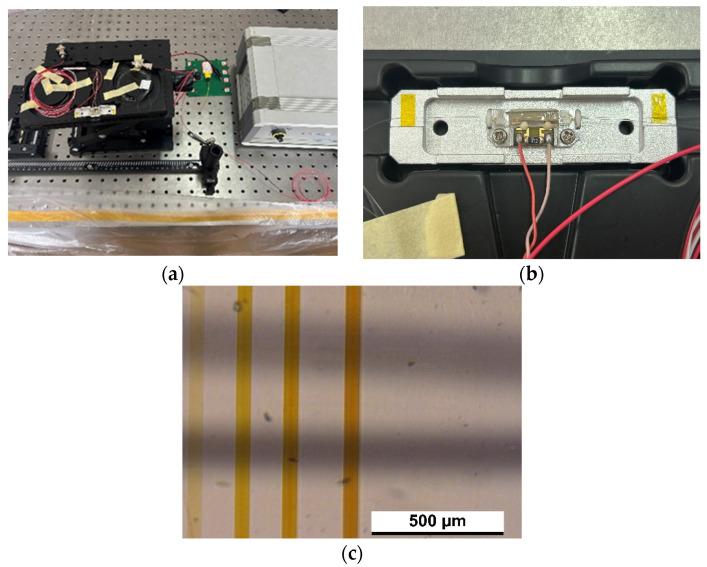
Electro-optic effect measurement. (**a**) Electric field measuring platform (complete device); (**b**) electrode contacts bonding; (**c**) electrode contacts bonding (microscopic image).

**Figure 13 sensors-24-05570-f013:**
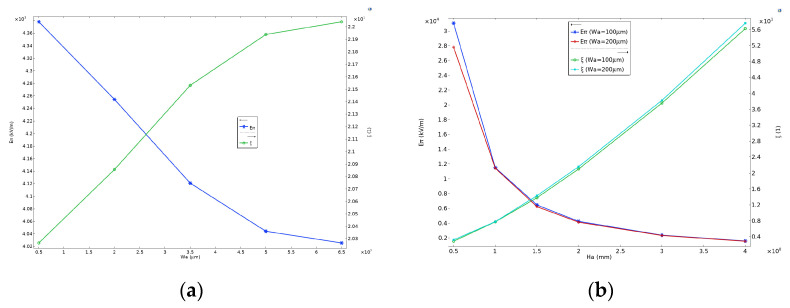
The half-wavelength electric field $E_{\pi }$ and the electric field gain value at the probe position ξ varying with the antenna width $W_{a}$ and the antenna height $H_{a}.$ (**a**) Relationship between the half-wavelength electric field and the electric field gain and the antenna width. (**b**) Relationship between the half-wavelength electric field and the electric field gain and the antenna height.

**Figure 14 sensors-24-05570-f014:**
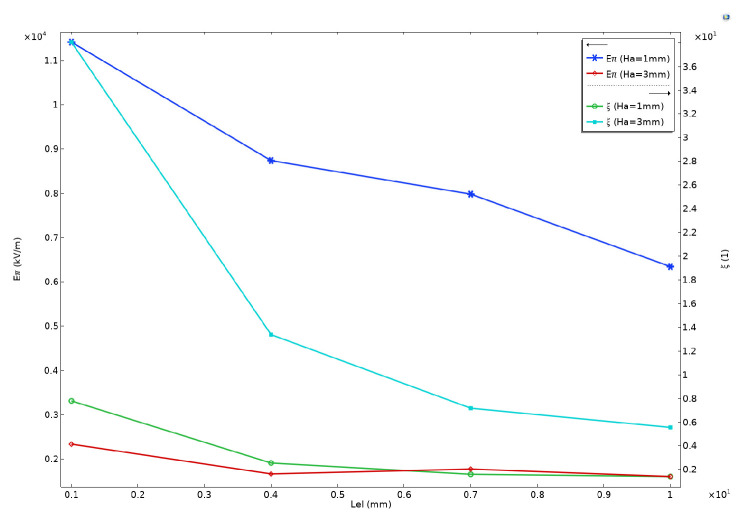
The half-wavelength electric field $E_{\pi }$ and the electric field gain value at the probe position ξ varying with the antenna length $L_{el}$ when $H_{a}=$1 mm and 3 mm.

**Figure 15 sensors-24-05570-f015:**
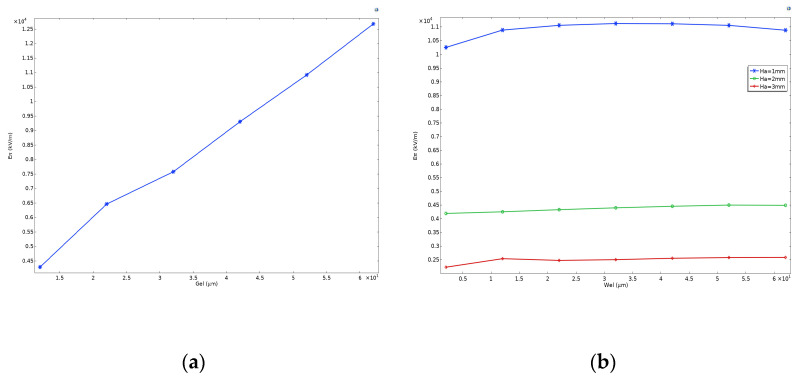
The half-wavelength electric field $E_{\pi }$ varying with the electrode width $W_{el}$ and the electrode spacing $G_{el}$. (**a**) Relationship between the half-wavelength electric field and the electric field gain and the antenna width when $H_{a}=$1 mm, 2 mm, and 3 mm. (**b**) Relationship between the half-wavelength electric field and the electrode spacing.

**Table 1 sensors-24-05570-t001:** Morphology of cladding-type waveguide end facet corresponding to different writing parameters.

repetition frequency: 1 kHz	
0.5 mm/s	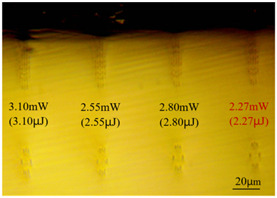
repetition frequency: 10 kHz	
0.5 mm/s	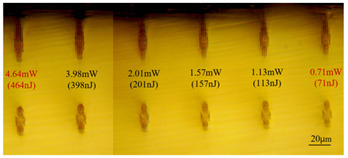
2 mm/s	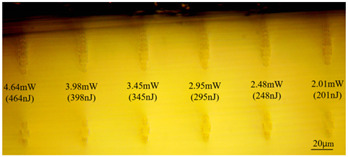
repetition frequency: 100 kHz	
0.5 mm/s	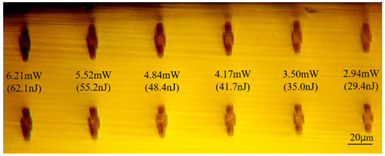
2 mm/s	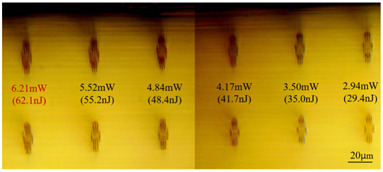

**Table 2 sensors-24-05570-t002:** The top-down morphology of the cladding-type waveguide corresponding to different writing parameters.

Repetition Frequency	Scan Speed	Depth	Average Power	Top View of Waveguide Topography
1 kHz	0.5 mm/s	60 μm	2.27 mw	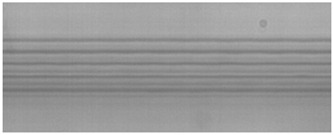
		110 μm		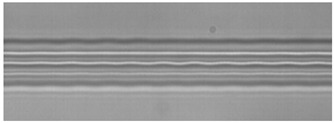
10 kHz	0.5 mm/s	60 μm	0.71 mw	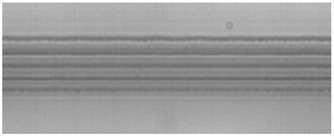
			44 mw	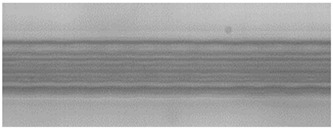
100 kHz	2 mm/s	60 μm	6.21 mw	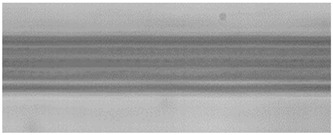

**Table 3 sensors-24-05570-t003:** Morphology of cladding waveguide end facets corresponding to different writing parameters.

repetition frequency: 100 kHz		
0.1 mm/s	2 mm/s	3 mm/s
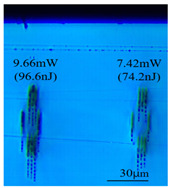	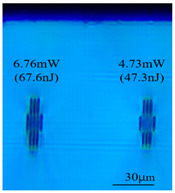	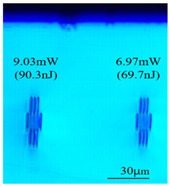
repetition frequency: 200 kHz		
0.1 mm/s	2 mm/s	3 mm/s
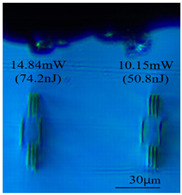	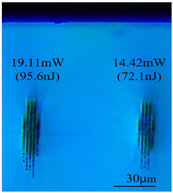	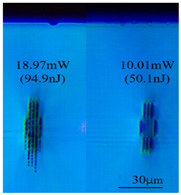

**Table 4 sensors-24-05570-t004:** Morphologies of partial refractive-index-increase-type waveguide written by the femtosecond laser.

$d_{Ι}$	End View	Top View
2 μm	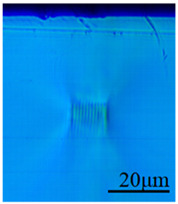	
1 μm	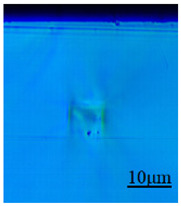	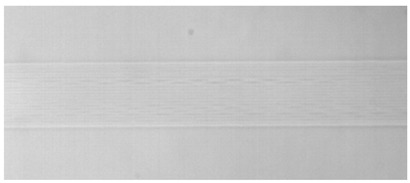

## Data Availability

Data are contained within the article.

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
