# Peer review of "Integrated Optical Waveguide Electric Field Sensors Based on Bismuth Germanate"

_sensors, 2024, doi:10.3390/s24175570_

Round 1

Reviewer 1 Report

Comments and Suggestions for Authors

Manuscript review: “Integrated Optical Waveguide Electric Field Sensors Based on Bismuth Germanate”, Sensors, MDPI, Basel

OVERALL VIEW: This manuscript intends to describe an experience of integrated optical waveguide for electric field sensor composed of bismuth germanate (Bi4Ge3O12, BGO), first by description of the experimental platform of femtosecond inscription on the BGO crystal and optimization of these experiment, and further, by the construction of antennas and sensors based on those crystals and finally – the evaluation of constructed sensors for the measurement of the electric field. However, the topic sounds interesting, the work on integrated optical waveguides for this kind of sensors in not pioneering and the manuscript needs to be carefully revised in order to be accepted in this journal, in my opinion.

DETAILLED REMARQUES: The proposed paper reveals the research on the optical waveguides constructed on the bismuth germanate crystals but the way of construction of this manuscript leaves an impression of technical report more than the scientific paper. This manuscript is written and prepared in very scholar manner, with the lack of relevant references in the field [such as Optical Mat. Ex., vol.5, Iss.2, pp.323-329 (2015) or Applied Optics, vol. 59, Iss.21, pp. 6237-6244 (2020)]. The introduction part is very long and it gives an impression of historical review more than the factual background to the described research. It lacks of bibliographic references focused on the research topic. The description of previous cited work on the form “MAIN AUTHOR and his colleagues” should be replaced by “MAIN AUTHOR et al.“. Also, the contribution of authors should be cited and not the Chinese Universities’ names.

From the edition and text preparation aspect, there is a mismatch of the numeration of paragraphs and sub-paragraphs (ex. line 246 – paragraph 3, line 366 – sub-paragraph number, line 520 –paragraph 4, etc.), also the tables and figures are not described in the correct way and there is a gap between some tables. The conclusion is very rudimentary and there is no discussion part evolved from presented results.

The quality of some presented images and figures is very low (ex. Fig.13, Fig. 14).

More specific comments and remarks:

1.       The address of the institution is not complete and does not permit to find the authors’ research structure.

2.       Could Authors explain the term “OVS” applied in the introduction part?

3.       What is the source of the bismuth germanate crystal, is it a commercial sample or the synthetic one? For the clarity of description, all the materials and equipment should be annotated as follow: name of equipment (company, country).

4.       What was the method of determination of losses of waveguide?

5.       On the Fig.10 the parameters to describe the conical dipole antenna (such as Ha, Wa, Gel, Wel) are not introduce nor explained and there is no equation bonding those parameters in sub-paragraph 4.3 (lines 604-650). Could Authors complete this part?

6.       The paragraph 3.1-3.4 should be rewrite in a comprehensive manner.

7.       In the Table 4, the scale of images and description is missing.

Comments on the Quality of English Language

ENGLISH SPELLING and EDITION: English spelling of this manuscript is correct and the grammar does not show a main issue. Yet, there are some edition problems with chemical formula, where the number should appear in the subscript and whole manuscript should we carefully checked for the mismatching of number of tables, citations (for instance in the line 94 the ref. [105]), figures and equations (in the equation 8, terms n2 and n3 are replaced). In general, the Greek and Roman terms used in text such as “in situ”, “via” should be written in italics.

Reviewer 2 Report

Comments and Suggestions for Authors

The research conducted in this paper has potential and deserves publication in the Sensors journal. However, I have the following comments which needs to be addressed before the final verdict:

1) The references should be cited properly. In line 94, reference 105 has been cited which appears after reference 9. Maybe its a typo-error. 

2) Provide the color bar to figures 6, 7 and 8.

3) The proper formatting of the paper is required. The figures are not inserted properly in the paper, for instance figure 16. 

4) In the introduction section, several statements are quoted without providing proper referencing to it. I suggest adding the latest literature to support the claims made in the introduction section. For instance: "With the advancement of integrated optics technology, electric field sensors based on 73 integrated optics have garnered widespread attention, typically utilizing optical wave- 74 guides integrated on planar substrates. "; "In recent years, optical electric field sensors have predominantly relied on bulk crys- 48 tal structures, employing single or double crystal configurations for electric field meas- 49 urement."  And many more. 

5) Add a characteristics table to compare the sensing performance of previously demonstrated electric field sensors with the one demonstrated in this work. 

6) I assume that the paper is unnecessarily big. I suggest removing some parts which are too vital (such as theory, detail on working mechanism, etc) to be added in the main text can go to supplementary data. And only present the main contribution of the author. 

7) The caption of Table 1 has some errors. Instead of "cladding-type waveguide end", it should be written as "cladding-type waveguide end-facet".

8) Did the author perform numerical simulations alongwith the experimental demonstration? Can the author show the E-field distribution in the proposed waveguide numerically?

Comments on the Quality of English Language

none. 

Round 2

Reviewer 2 Report

Comments and Suggestions for Authors

The author has not fully answered my concerns. I am unsatisfied with the author's response. I suggest responding to all the points and submitting the report. I asked 8 questions, whereas the author responded with only 5. 

Comments on the Quality of English Language

None. 

Round 3

Reviewer 2 Report

Comments and Suggestions for Authors

I am willing to accept the paper in its current form. 

Comments on the Quality of English Language

none.